# The State of the Art of Molecular Fecal Investigations for *Helicobacter pylori* (*H. pylori*) Antibiotic Resistances

**DOI:** 10.3390/ijms24054361

**Published:** 2023-02-22

**Authors:** Francesca Celiberto, Giuseppe Losurdo, Maria Pricci, Bruna Girardi, Angela Marotti, Alfredo Di Leo, Enzo Ierardi

**Affiliations:** 1Section of Gastroenterology, Department of Precision and Regenerative Medicine and Ionian Area, University of Bari, 70124 Bari, Italy; 2Course in Organs and Tissues Transplantation and Cellular Therapies, Department of Precision Medicine Jonic Area, University “Aldo Moro” of Bari, 70124 Bari, Italy; 3THD SpA, 42015 Correggio, Italy

**Keywords:** *Helicobacter pylori*, stools, antibiotic resistance, therapy, eradication, genotypic resistance

## Abstract

A new paradigm shift for the treatment of *Helicobacter pylori* (*H. pylori*) infection would be timely due to a progressive increase in antibiotic resistance. Such a shift in the perspective of the *H. pylori* approach should include the preliminary assessment of antibiotic resistance. However, the availability of sensitivity tests is not widespread and the guidelines have always indicated empirical treatments without taking into account the need to make sensitivity tests accessible, i.e., the necessary starting point for improving results in different geographical areas. Currently, the traditional tools for this purpose (culture) are based on performing an invasive investigation (endoscopy) and often involve technical difficulties; thus, they were only confined to the settings where multiple attempts at eradication have failed. In contrast, genotypic resistance testing of fecal samples using molecular biology methods is much less invasive and more acceptable to patients. The purpose of this review is to update the state of the art of molecular fecal susceptibility testing for the management of this infection and to extensively discuss the potential benefits of their large-scale deployment, i.e., novel pharmacological opportunities.

## 1. Introduction

It is well known that *Helicobacter pylori (H. pylori)* represents the infectious agent of chronic active gastritis, peptic ulcer disease, gastric carcinoma and MALT lymphoma as well as of some extra-gastric disorders, i.e., iron deficiency anemia and idiopathic thrombocytopenia. Its role in these conditions has brought about the recommendation that this bacterium should be eradicated whenever possible [1]. Nevertheless, the management of the infection at present is far from an optimal solution, since it displays some complex evidences: (i) infection is still widespread [2]; (ii) a large and puzzling number of empiric therapeutic schedules have never been proposed for any other infection [3]; (iii) the same regimen has given exciting and/or disappointing results at the same time [4]; (iv) currently, no available therapy has demonstrated a 100% successful rate [5].

It is unquestionable that these problems are mainly related to the progressive development of antibiotic resistances [6]. Undeniably, this matter has been growing all the time, thus inducing a continuous decline in the effectiveness of conventional therapies despite a variety of published treatment guidelines. These circumstances have led to the use of empirical treatments originally developed on the bases of clinical experiments and, even, breakdowns. Consequently, most of the guidelines have been based on increasingly less effective therapies, thus suggesting still a feasible use of antibiotics, such as clarithromycin, even after the spread of resistance had rendered them largely ineffective [7]. The same guidelines have never taken into account the need to make the sensitivity tests more accessible, i.e., the necessary starting point to improve the results in the various geographical areas, where it is known that the levels of resistance to various antibiotics are very different from zone to zone. It is evident that these concerns underline the importance of applying an antimicrobial management with regard to the treatment of *H. pylori*. In this context, the main problem has been the unavailability in first line of sensitivity tests, which enhanced the effects of microbial resistances on current therapy failures [8]. On these bases, it would be advantageous that next guidelines are adjusted according to susceptibility-based treatments rather than empirical clinical trials, meta-analyses based on them, and expert opinions [1,9].

Therefore, the purpose of this review is to testify the effectiveness and feasibility of current antibiotic resistance investigations and update the state of the art of molecular fecal susceptibility tests for the management of *H. pylori* infection, since they might be the most promising ones because of their noninvasive peculiarity and short time requiring results.

## 2. *H. pylori* Susceptibility Tests

The first susceptibility test was conventionally based on culture and antibiogram in *H. pylori* isolates (phenotypic resistance detection), even if it is recommended by current guidelines only after repeated treatment failures [1]. Indeed, it is almost impossible to use this method for first-line treatment selection, since a relatively high rate of false negative findings often resulting in a low sensitivity has weighed on the use of the test on a large scale. This intricacy is mainly due to the need to create and maintain a micro-aerophilic environment, in which the bacterium can grow. Other factors, that have prevented the widespread diffusion of *H. pylori* culture so far. are the following methodology-related issues: number of gastric biopsies, time-consuming endoscopic procedures, conditions and interval of biopsy samples transport, laboratory characteristics, long and unpredictable time needed to obtain the result of the investigation [10]. It is noteworthy that culture does not detect the *H. pylori* hetero-resistant state, i.e., the simultaneous presence of susceptible and resistant strains [11].

As an alternative to bacterium culture and antibiogram, real time polymerase chain reaction (RT-PCR)-based techniques have been developed (genotypic resistance detection) [12]. They are based on the principle of amplifying and detecting the point mutations responsible for antibiotic resistance of *H. pylori* DNA isolated from gastric biopsy samples. These culture-free approaches are accurate in revealing minimal traces of genotypic resistant strains as well as in finding out hetero-resistant status. Furthermore, the ability to evaluate resistant mutant genotypes by PCR not only on fresh samples, but also on archived paraffin-embedded biopsy specimens, which has been shown to provide an equally reliable substrate for the analysis of DNA as fresh material, has further emphasized the utility of these methods [13].

The pros and cons of the two methods (culture and RT-PCR on gastric biopsy samples) have been excellently summarized in a 2010 post hoc study enrolling 146 *H. pylori* positive patients and comparing phenotypic and genotypic methods for clarithromycin resistance analysis. Culture revealed an overall prevalence of phenotypic clarithromycin resistance significantly lower than RT-PCR with an agreement of 71.2% between the two techniques [14]. Three main factors may be invoked to explain the lack of a full agreement between the two methods: (i) the relatively low sensitivity of the phenotypic investigation, (ii) its lack of detection of hetero-resistance, (iii) the possibility that culture may identify resistant strains carrying rare and new-fangled point mutations that are different from commonly tested ones [11,14].

Based on what has been reported, molecular tests offer unquestionably advantages and guarantees of feasibility when compared to culture, even if they are not used in clinical practice. It is presumable that the need for an invasive endoscopic procedure has been the most significant limit to their diffusion. Therefore, a successive step has been represented by an attempt to overcome this drawback by pointing out an extensive and fitting analysis.

### 2.1. Molecular Fecal H. pylori Susceptibility Investigations: Early Realities

At first, a pioneering study in 1996 showed that it was possible to isolate *H. pylori* DNA from stool samples. This finding was confirmed by another evidence only after seven years [15]. A further advance in this topic was represented by the possibility of detecting the point mutations that confer antibiotic resistances in fecal samples of bacterial DNA. In this regard, Table 1 reports the main clarithromycin sensitivity studies performed on *H. pylori* fecal DNA by RT-PCR starting from 2003 [16,17,18,19,20,21,22,23,24,25,26]. The most interesting aspect which emerged from these studies was that all showed high sensitivity and most of them also showed high specificity, when compared with culture and/or RT-PCR of gastric biopsies. Interestingly, the method appeared to be reliable even in diagnosing infection when compared with the most commonly used non-invasive diagnostic methods (^13^C-urea breath test and stool antigen detection).

One of the first commercial non-invasive investigations using fecal RT-PCR (*H. pylori* ClariRes assay, Ingenetix, Vienna, Austria) tested the A2142G mutation for clarithromycin resistance and was used in a pediatric population (143 children). Its main limit was constituted by the presence of the other two main mutations responsible for resistance to this antibiotic in Western countries (A2143G and A2142C), whose search was not foreseen by this test [27].

Later, another commercial molecular test was developed, i.e., Genotype Helico-DR (Hain Lifescience GmbH, Nehren, Germany). It allowed the detection of the molecular resistances of *H. pylori* to both clarithromycin and fluoroquinolones, identifying both the most common point mutations for the resistance to clarithromycin (A2146C, A2146G and A2147G) and to the fluoroquinolones, i.e., the mutations of the gyrA gene located at positions 87 (N87K) and 91 (D91N, D91G, D91Y) [28,29]. This investigation was initially used on tissue samples and, only later, applied to *H. pylori* DNA isolated from fecal samples [29]. The limitation demonstrated by this test was the finding of an unexpected low agreement between the detection of resistance to clarithromycin and fluoroquinolones on stool and gastric biopsy samples, respectively.

Simultaneously with the use of the HelicoDR test on stool samples, a new RT-PCR method (THD Fecal Test, THD S. p. A., Correggio, Italy) was preliminarily tested to study clarithromycin resistance mutations in stool bacterial DNA [30]. The procedure showed full agreement between the results obtained in tissue and stool in 52 consecutive patients at the first diagnosis of infection. The A2143G mutation was found in ten (19.2%), A2142G in four (7.7%) and A2142C in five (9.6%) patients with an overall clarithromycin resistance rate of 23% in a Southern Italian population. In order to better understand the diagnostic reliability of the test, the results of a preliminary “in vitro” experiment are of interest. This experiment demonstrated that the presence of components of fecal material, such as macromolecules and fibers, makes necessary that a certain amount of colony forming units (CFU)/mL are present to obtain a positive result from the isolation of bacterial DNA. In detail, a clear positivity was achieved by a concentration of 1.5 × 10 CFU/mL of pure bacteria and of 1.5 × 10^3^ CFU/mL of a mixture of micro-organisms and feces. Therefore, fecal test results for *H. pylori* DNA may be influenced by a cut-off value for bacterial concentration in feces.

On these bases, it is presumable that the lack of agreement between the results on the stool and the gastric biopsy samples observed with Gene Helico-DR could be due specifically to the fact that a cut-off value for bacterial concentration in feces had not been evaluated for this investigation.

### 2.2. Molecular Fecal H. pylori Susceptibility Investigations: Evolution of Research to Date

Kovacheva-Slavova et al. performed a study in 2021 enrolling 50 patients with dyspeptic symptoms aging 46.46 ± 15.10 year using a molecular test based on RT-PCR in fresh fecal samples (VIASURE *H. pylori* real-time PCR Detection Kit; CerTest Biotec S.L. Zaragoza, Spain). A stool antigen test was used as gold standard. They identified *H. pylori* infection in 24 patients (48.00%). Clarithromycin resistance was observed in seven of them (29.17%). None of the patients had been treated before. The molecular test showed 85.71% sensitivity and 100% specificity, with a diagnostic accuracy of 92.00% [22]. The study confirmed that this molecular test could be beneficial for its high accuracy and clarithromycin resistance. Its assessment could improve the outcome of eradication therapy.

Marrero Rolon et al. developed a PCR assay using a customized extraction kit in 2021 (Mayo MicroLab Maxwell high-throughput fecal DNA purification kit chemistry; Promega, Madison, WI, USA) that employed methods to enhance inhibitor removal and maximize DNA extraction from stool samples for the simultaneous detection of *H. pylori* and genotypic markers of clarithromycin resistance (A2143G, A2142G, and A2142C) directly from stool specimens. The test resulted in 88.6% and 92.8% sensitivity in the validation and clinical study sets, respectively. A high value of specificity was observed (97%). Sequencing confirmed correct detection of clarithromycin resistance-associated mutations in all positive validation samples. The gold standard in this study was culture [23]. A set of 223 antigen-positive stool samples was tested and retrospective medical record review performed to define the clinical utility. The clarithromycin-based triple therapy success was very low in the presence of resistance detection by PCR (41%) when compared to the absence of resistance finding (70%; *p* = 0.03).

A further prospective study on GenoType Helico DR assay was performed by Brennan et al. in 2016 in a study population of 616 subjects. Genetic identification of *H. pylori* and its resistance to clarithromycin and fluoroquinolones was performed on stool samples from patients with campylobacter-like organism positive endoscopy (389) and UBT-positive patients (227). A multiplex amplification of DNA regions of interest was performed using a combination of the biotinylated primers supplied in the GenoType HelicoDR kit (Hain Lifescience GmbH, Nehren, Germany) and the Hotstart Taq DNA polymerase kit (Quiagen, Hilden, Germany). PCR products were reverse hybridized to DNA strips containing probes for gene regions of interest. According to conventional techniques, the strips were analyzed for the presence of a conjugate control band (to indicate successful conjugate binding and substrate reaction), an amplification control band (to indicate a successful amplification reaction), a *H. pylori* control band (to document the presence of a *H. pylori* strain) and gene locus control bands for 23S (positions 2146 and 2147) and gyrA (codons 87 and 91) in order to indicate the successful detection of the gene regions of interest for clarithromycin and fluroquinolone resistance, respectively. In addition, the strips were analyzed for the presence of wild type and/or mutation bands. The assay was reported to be efficient at detecting mutations predictive of antibiotic resistance when applied to *H. pylori* cultures or gastric biopsy specimens, with a sensitivity and specificity of 94–100% and 86–99% for detecting clarithromycin resistance and 83–87% and 95–98.5% for detecting fluoroquinolone resistance, respectively. The gold standard of this study was PCR on gastric biopsy samples [24]. Nevertheless, authors emphasized that the GenoType HelicoDR assay was not suitable for the accurate detection of antibiotic resistance-mediating mutations using stool samples from *H. pylori*-infected patients and alternative PCR or DNA sequencing-based methods could show a better outcome. As reported above the limit of this test may have been due to the fact that a cut-off value for bacterial concentration in feces was been evaluated for this investigation.

In 2020, Pichon et al. [25] described a real-time PCR (*H. pylori* ClariR) assay that allowed the amplification of samples from stool. The raw data were analyzed by using a fully automated analysis program (Amplidiag Analyzer) that accelerated this process, obtaining the same results (Amplidiag *H. pylori* + ClariR, Mobidiag, Espoo, Finland). The test provided results for the detection of *H. pylori* and for the detection of mutations conferring clarithromycin resistance (without distinction between the mutations). A prospective, multicenter study enclosed 1200 adult patients who underwent gastroduodenal endoscopy with gastric biopsy sampling and were naive for eradication treatment. The results were compared with those of culture/E test. Quadruplex real-time PCR was performed on two gastric biopsy samples (from the antrum and corpus) in order to detect the *H. pylori* glmM gene and mutations in the 23S rRNA genes conferring clarithromycin resistance. The sensitivity and specificity of the detection of *H. pylori* were 96.3% (95% confidence interval [CI], 92 to 98%) and 98.7% (95% CI, 97 to 99%), respectively. Positive and negative predictive values were found to be 92.2% (95% CI, 92 to 98%) and 99.3% (95% CI, 98 to 99%), respectively. In this cohort, 160 patients (14.7%) were found to be infected (positive by culture and/or PCR). The sensitivity and specificity for detecting resistance to clarithromycin were 100% (95% CI, 88 to 100%) and 98.4% (95% CI, 94 to 99%), respectively [25].

Another nested polymerase chain reaction-quenching probe (Nested PCR-QP) with a novel technique was pointed out by Kakiuchi et al. in 2020 in order to analyze 23S rRNA genetic mutations (A2142C, A2142G, and A2143G) that were associated with clarithromycin resistance in *H. pylori* [31]. In a sample of 57 *H. pylori*-positive subjects, a clarithromycin rate of resistance of 49% was observed.

A further study demonstrated THD fecal test diagnostic accuracy when compared to ^13^C urea breath test. Two hundred and ninety participants completed the study. The THD fecal test showed the following results: sensitivity 90.2% (CI: 84.2–96.3%), specificity 98.5% (CI:96.8–100%), PPV 96.5% (CI: 92.6–100%), NPV 95.6% (CI: 92.8–98.4%), accuracy 95.9% (CI: 93.6–98.2%), positive LR 59.5(CI: 19.3–183.4), negative LR 0.10 (CI: 0.05–0.18). Out of 83 infected participants identified with the THD fecal test, 34 (41.0%) had bacterial genotypic changes consistent with antibiotic-resistant *H. pylori* infection. In detail, 27 subjects (32.5%) demonstrated bacterial strains resistant to clarithromycin, 3 (3.6%) to levofloxacin, and 4 (4.8%) to both antibiotics [26].

## 3. Use of Molecular Tests in Clinical Practice

Since 2007, some studies evaluating guided versus empirical treatment after PCR resistance detection were conducted (Table 2).

The first attempt was preliminarily performed in Japan by Furuta et al. with a surprising result. Indeed, susceptibility guided treatment success was lower than empirical treatment (75% vs. 84.4% at intention to treat—ITT). This study, however, was strongly limited by the very small number of enrolled patients (four) in the group of tailored therapy. Indeed, when the same authors carried out a second investigation on a larger sample (300 patients), a high success rate was found in guided (96%) when compared with the empirical therapy group (70%) at ITT [32].

A similar study was conducted in 2015 by Dong et al. from China. The results did not differ much from Japan, i.e., with an outcome of an eradication rate of 91.1% for guided versus 73.3% for empirical treatment at ITT [33].

Successively, in 2018, Liou et al. reported the results of two studies from Taiwan in the same paper. Resistance-associated mutations in 23S ribosomal RNA (clarithromycin) or gyrase A (fluroquinolones) were identified by polymerase chain reaction with direct sequencing. The differences between tailored and empirical treatment were evident in both preliminary (81% versus 60% at ITT in 41 patients) and final experiment (78% versus 72.2% at ITT in 410 patients) [34]. Despite this satisfactory outcome, authors expressed reservations regarding tailored therapy accessibility, cost, and patient preference.

A further experiment from China was performed by Fan et al. In this study, PCR investigation was supplemented by sequencing. The study provided a comparison between clarithromycin containing quadruple therapy versus tailored quadruple therapy. AT ITT, the difference between guided (77.8% in 270 subjects) and empirical (65.3%; 274 subjects) treatment success rate was evident even if overall eradication percentage was almost low [35]. Nevertheless, this value markedly increased when per protocol analysis was performed: 86.4% versus 70.2%.

The most recent experiment is that from Delchier et al., who used GenoType HelicoDR in order to compare tailored PCR-guided and empirical triple therapy. This French multicenter prospective open-label randomized study enclosed 207 subjects in guided therapy and 208 in empirical therapy. The results confirmed the superiority of guided (85.5%) when compared to empirical therapy (73.1%) [36].

Finally, Ma et al. recently reported a systematic review and meta-analysis about tailored therapy for *H. pylori*, enclosing both PCR and culture-based regimens. When the results were limited to PCR-guided therapy, a better outcome for tailored therapy was found, with an odds ratio of 1.24 at ITT (95% CI 1.12–1.36) [37].

## 4. Discussion

The treatment of *H. pylori* infection shows some critical issues, due to the fact that the regimens proposed over the years are losing their effectiveness because of the development of antibiotic resistance. At the same time, in the last 20 years, the arsenal of available drugs has not changed in number, but the combinations have simply been flourished in order to improve their outcome. On these bases, it would be desirable that the treatment of the infection follows the criteria of “precision medicine” and that a personalized treatment can be achieved.

The availability of susceptibility testing for *H. pylori*, therefore, may change the management of this infection, at least conceptually, in agreement with that of most infectious diseases. This potential advancement could modernize the current management, which basically consists of opinion-based recommendations, with a progression towards a susceptibility-based approach according to the current principles of antibiotic management. This will not be simple or fast for several reasons. Indeed, we currently have treatments that can provide a 90% success rate and are recommended by current guidelines. This could lead to the development of guidelines, which, at least in the near future, will likely continue to be inherent in a context, where the current principles of antibiotic management are not quick-witted and potential controversies may easily arise.

An obvious objection that can be raised to the current therapeutic indications is that suggested regimens require the daily intake of a large number of tablets, respectively, fourteen for bismuth containing quadruple therapy, and eight for the concomitant regimen [1,9]. It is clear that this aspect can negatively influence the patient compliance. It is presumable that a complete adherence may be obtained from patients who may be motivated by the presence of major diseases, such as MALT lymphoma, a family history of gastric cancer or peptic ulcer, particularly if complicated by bleeding episodes. However, dyspeptic or asymptomatic subjects may not be guided by similar motivations, given that, in most cases, the eradication of the bacterium is not accompanied by a clear clinical benefit [38]. Indeed, the incomplete adherence to an antibiotic therapy may be an important cause of resistance emergence since sub-inhibitory concentrations could stimulate the selection of resistant mutants. In this regard, bismuth containing quadruple therapy encompasses the use of tetracycline, which currently shows very low resistance rates in Europe. However, tetracycline resistance rates of 19% have already been reported in Asia [39]. Therefore, as already occurred for other therapeutic regimens as triple therapy (Table 3) [40,41,42,43,44], in the future, there is the truthful risk of an increase in resistance to this antibiotic due to its large-scale use as well as patient incomplete adherence to its intake [45].

Conversely, concomitant therapy involves the use of three conventional antibiotics with the obvious aim of overcoming resistance to each individual drug by the overall combined effect of the regimen. This strategy, therefore, is based on hypothetical assessments rather than real susceptibility data and, presumably, encompasses the use of more antibiotics than needed.

Conversely, triple therapy containing amoxicillin and clarithromycin is currently no longer recommended in areas with a 30% of clarithromycin resistance, where it has been proven to be ineffective in 40–50% of patients. The progressive reduction in the efficacy of this therapy in the decades 1997–2017 is summarized in Table 3. Nevertheless, despite its significant ineffectiveness, it could still be used for clarithromycin susceptible strains, if we have the possibility to evaluate this feature before prescribing an eradication treatment. Therefore, the availability of sensitivity tests might bring this and other issues into focus and could address their solution.

It is, therefore, possible that the high efficacy rate of currently recommended therapies could lead to an excessive and superfluous consumption of antibiotics and that this aspect could reduce their efficacy in the future, favoring the development of increasingly resistant strains.

Based on what has been reported above, it would be appropriate that susceptibility tests are available, which do not require invasive investigations and allow the immediate knowledge of the result. For these reasons, several attempts have been made at developing genotypic investigations of susceptibility on fecal samples, which potentially have all the required requisites. Molecular methods have few limitations since they are based on the amplification of small amounts of bacterial DNA and, therefore, are very sensitive. However, there are some rare mutations (such as A2115G, G2141A, and A2144T for clarithromycin), which are not detected by commercially available kits [46]. Furthermore, in some cases, melting curve-based methods may detect mutations that are neutral and do not confer any resistance, thus causing false positives [47].

At present, several studies have confirmed the reliability of these tests at least with regard to the evaluation of resistances to clarithromycin and fluoroquinolones [24,48], while the results concerning metronidazole have given controversial results and need to be further improved [49]. The data regarding amoxicillin and tetracycline are still insufficient and need to be validated by experiments on large samples. Other antibiotics that have shown efficacy in empirical regimens (rifabutin, doxycycline, minocycline) should be considered in the development of molecular tests of susceptibility for second-third line regimens [50,51,52]. Similarly, antibiotics that have been shown in vitro to be effective in strains with multiple resistances (tigecycline) could be considered and tested for rescue-therapies [53].

A separate discussion deserves the use of furazolidone for the treatment of *H. pylori* infection for the relevant ethical concerns arising with its use. This is an antibiotic that was used in the 1980s for parasitic infections. Studies were published in the 1990s that raised several concerns about this drug for its potential carcinogenicity. Therefore, the FDA withdrew its approval in March 2005. At the same time, the European Medicines Agency (EMEA; the equivalent of the FDA in the European Union) banned the drug in Europe. Therefore, its use is limited to countries outside the United States and the European Union and carries important risks for patients, who should at least be informed about them [54].

In order to optimize therapeutic regimens, it should be also considered that proton pump inhibitors (PPIs) differ markedly in terms of antisecretory activity and that intragastric pH control is a critical determinant of the success of a curative schedule for *H. pylori* infection. Currently, even the comparative studies of PPIs as adjuvants of therapy have been conducted with a certain superficiality. In fact, these are substances with largely different antisecretory efficacy. For example, 40 mg of pantoprazole has an antisecretory efficacy equal to 9 mg of omeprazole, while 20 mg of vonoprazan has an effect greater than 70 mg of omeprazole [55]. Therefore, it is evident that a legitimate comparison of therapeutic regimens would require the use of antisecretory drugs of equivalent potency.

Based on what has been reported above, about the possibilities and limits of the use of genotypic methods for resistances, we would state once more that they are currently reliable only for the evaluation of resistance to clarithromycin and fluroquinolones [24,48]. Therefore, we conclude that genotypic techniques still require further development so that they give results that are completely comparable to the phenotypic method. Despite these concerns representing an undeniable reality, the relevant detail remains that culture method is not feasible as a front-line technique for guided therapy.

## 5. Conclusions

A wide diffusion of sensitivity tests on fecal samples, even if currently validated and usable only for clarithromycin and fluoroquinolone resistance detection, could accelerate the process of simplifying and personalizing the therapeutic choice of *H. pylori* infection, thus inducing novel pharmacological chances even with the use of old drugs. In this regard, it is important to consider that none of the recommended therapies were optimized before their approval and no limit for an acceptable cure rate was established as relevant part of the approval. “Packed-up” therapies, often containing components that are difficult to find, limit the possibility of their “personalization” in terms of drug, dosage, and duration. Finally, the possibility of using susceptibility tests before prescribing an eradication treatment, albeit at the moment in a field limited to two classes of antibiotics, could lead to the following results:a.Regimens that contain unnecessary drugs could be no longer approved or used in order for them to contribute to an increase in resistances;b.Doubts about the dosages and duration of the therapies could be eliminated;c.Comparisons of PPIs should provide the relative antisecretory potency of the drugs necessary for their therapeutic efficacy.

Indeed, a wise use of antibiotics is framed within the principles of antibiotics stewardship which, in the case of *H. pylori*, should rely on using therapies that are proved to be highly effective locally, performing a test-of-cure, and applying such data to confirm local effectiveness and share the results in the medical community [56].

## Figures and Tables

**Table 1 ijms-24-04361-t001:** Studies on *H. pylori* stool DNA by real time polymerase chain reaction (RT-PCR) for diagnosis of infection and/or antibiotic susceptibility (clarithromycin).

Author, Year	Gold Standard	Sensitivity (%)	Specificity (%)	Clarithromycin Resistance (%)
Marrero, 2021 [23]	Culture	89	97	39
Kovacheva, 2021 [22] *	Stool antigen	85.7	100	29.2
Pichon, 2020 [25]	Culture	96.3	98.7	NA
Iannone, 2018 [26] *	13C-urea breath test	90.2	98.5	32.5
Brennan, 2016 [24]	PCR on gastric biopsy	94	100	83
Scaletsky, 2011 [21]	Culture	94	100	26.7
Noguchi, 2007 [20]	Culture	NA	NA	20.4
Rimbara, 2009 [19]	Culture	96.6	91.3	13.3
Booka, 2005 [18]	Culture	NA	NA	31
Schabereiter, 2004 [17]	Culture	98	98	24.4
Fontana, 2003 [16]	Culture	100	100	1.6

* Diagnosis plus susceptibility; NA: not assessed.

**Table 2 ijms-24-04361-t002:** Studies evaluating guided versus empirical treatment after PCR resistance detection.

Author	Susceptibility-Guided Treatment Success: % (n/N) at ITT *	Empirical Treatment Success: % (n/N) at ITT *
Furuta, 2007 [32]	75% (3/4)	84.4% (27/32)
Furuta, 2007 [32]	96% (144/150)	70% (105/150)
Dong, 2015 [33]	91.1% (41/45)	73,3% (33/45)
Liou, 2018 [34]	81% (17/21)	60% (15/20)
Liou, 2018 [34]	78% (164/205)	72.2% (162/205)
Fan, 2019 [35]	77.8% (210/270)	65.3% (179/274)
Delchier, 2020 [36]	85.5% (177/207)	73.1% (152/208)

* ITT: intention to treat.

**Table 3 ijms-24-04361-t003:** Reduction in the efficacy of triple therapy in the decades 1997–2017.

Author, Year	Regimen (Duration)	Success Rate (%)
Wurzer H, 1997 [40]	OAC (10 days)	91.0
Bazzoli, 1998 [41]	OMC (7 days)	93.8
Venerito, 2013 [43]	OAC (7 days)	68.9
Kekilli, 2016 [42]	OAC (10 days)	64.7
Chang, 2017 [44]	OAC (10 days)	64.3

O: omeprazole 20 mg b. i. d.; A: amoxicillin 1 g b. i. d.; C: clarithromycin 250–500 mg b. i. d.

## Data Availability

Not applicable.

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
