# Peer review of "The State of the Art of Molecular Fecal Investigations for Helicobacter pylori (H. pylori) Antibiotic Resistances"

_ijms, 2023, doi:10.3390/ijms24054361_

Round 1
Reviewer 1 Report
I believe that the authors gave an excellent presentation of the current possibilities regarding the molecular analysis of stool for H. pylori infection. However, I would like to make a few comments: 1) in the Introduction, the authors refer to the Maastricht V guidelines. Given that the Maastricht VI guidelines were published in the meantime, I think that the ideas in that part of the text should be based on newer guidelines. 2) table 3 should be excluded from the conclusion and the entire text refferring to the triple therapy should be moved to the discussion.Author Response
I believe that the authors gave an excellent presentation of the current possibilities regarding the molecular analysis of stool for H. pylori infection. However, I would like to make a few comments: 1) in the Introduction, the authors refer to the Maastricht V guidelines. Given that the Maastricht VI guidelines were published in the meantime, I think that the ideas in that part of the text should be based on newer guidelines.
We thank the reviewer for the kind appreciation. In agreement with his suggestion, we referred to the European guidelines Maastricht VI instead of Maastricht V.
2) table 3 should be excluded from the conclusion and the entire text referring to the triple therapy should be moved to the discussion.
In agreement with the reviewer’s suggestion, we moved all the text referring to the triple therapy and the related table in the body of the Discussion and deleted it in Conclusions.
Reviewer 2 Report
- H. pylori should be stated with italic font through the manuscript (title).
- In introduction, the authors should be highlighted the importance of applying the antimicrobial stewardship regarding the treatment of H. pylori.
- the sentence "responsible for antibiotic resistance of H. pylori DNA isolated". "of" should not be italic font.
please give more attention regarding grammatical mistakes as well as typos errors through the text.
- the authors should discuss about limitation of molecular methods i.e. evaluation of point mutations and its association with result of drug-susceptibility testing test.
- Conclusion should be objective with further perspectives.
- Finally, I believed that molecular methods have low sensitivity compared than phenotypic method (as gold standard) for detection of H. pylori-related antibiotic resistance.
Author Response
- H. pylori should be stated with italic font through the manuscript (title).
We apologize for the typo and have performed the suggested correction.
- In introduction, the authors should be highlighted the importance of applying the antimicrobial stewardship regarding the treatment of H. pylori.
The purpose of this review was precisely to underline that the availability of susceptibility-guided therapies represent the future of treatment of H. pylori infection. In the revised manuscript, we have made a further effort to underline this aspect. Additionally, we added a paragraph on antibiotic stewardship in the conclusions paragraph.
- the sentence "responsible for antibiotic resistance of H. pylori DNA isolated". "of" should not be italic font.
We apologize for the typo and have performed the suggested correction.
- please give more attention regarding grammatical mistakes as well as typos errors through the text.
We apologize for the typo and some grammatical mistake and have performed the appropriate corrections.
- the authors should discuss about limitation of molecular methods, i.e. evaluation of point mutations and its association with result of drug-susceptibility testing test.
We reported in the Discussion two paragraphs about both technical problems and current suitability of genotypic methods, respectively:
- “Molecular methods have few limitations since, based on the amplification of small amounts of bacterial DNA, they are very sensitive. However, it should be mentioned that there are some rare mutations (such as A2115G, G2141A and A2144T for clarithromycin), which are not detected by commercially available kit (reference 46 of revised manuscript). Further, in some cases, melting curve based methods may detect mutations that are neuter and do not confer any resistance, thus causing false positives (reference no. 47 of revised manuscript)”.
- “Based on what above reported about the possibilities and limits of the use of genotypic methods for resistances, we would state once more that they are currently reliable only for the evaluation of resistance to clarithromycin and fluroquinolones (references no. 29 and 48 of revised manuscript). Therefore, we can conclude that genotypic techniques still require further development so that they give results that are completely comparable to the phenotypic method. Despite these concerns represent an undeniable reality, the relevant detail remains that culture method is not feasible as a front-line technique for guided therapy”.
- Conclusion should be objective with further perspectives.
We discussed more in depth possible perspectives in the Conclusion and Discussion paragraphs. Moreover, some changes were made to the concluding remarks to accommodate this reviewer's request.
- Finally, I believed that molecular methods have low sensitivity compared than phenotypic method (as gold standard) for detection of H. pylori-related antibiotic resistance.
While showing great respect for the reviewer's opinion, in this manuscript we have reported data from the literature regarding the sensitivity of molecular methods compared with culture as the gold standard. Furthermore, we have provided evidence that the phenotypic method is not feasible as a front-line technique for guided treatment. For this reason, all guidelines suggest it only for rescue therapy. On the other hand, we also underlined that the genotypic method on fecal samples is currently reliable only for the evaluation of resistance to clarithromycin and fluroquinolones. Therefore, we believe that our discussion follows the literature data and leads to objective conclusions in order to indicate a potential solution to a problem that has not been solved.
Round 2
Reviewer 2 Report
Well revised.